# Clinical Evaluation of Bilateral Multiple Gingival Recession Treatment with Autogenous Connective Tissue Graft Associated with Low-Level Laser Therapy

**DOI:** 10.3390/jcm12062349

**Published:** 2023-03-17

**Authors:** Ricardo Roberto de Souza Fonseca, Camila Pantoja Silva, Beatriz Leal de Senna Sastre, Erich Brito Tanaka, Tábata Resque Beckmann Carvalho, Paula Gabriela Faciola Pessôa de Oliveira, Silvio Augusto Fernandes de Menezes, Rogério Valois Laurentino, Renata Pimentel de Oliveira, Roberta Pimentel de Oliveira, Andréa Dias Neves Lago, Luiz Fernando Almeida Machado

**Affiliations:** 1Biology of Infectious and Parasitic Agents Post-Graduate Program, Federal University of Pará, Belém 66075-110, PA, Brazil; 2Virology Laboratory, Institute of Biological Sciences, Federal University of Pará, Belém 66075-110, PA, Brazil; 3School of Dentistry, University Center of State of Pará, Belém 66060-575, PA, Brazil; 4School of Dentistry, Federal University of Pará, Belém 66075-110, PA, Brazil; 5School of Dentistry, Federal University of Maranhão, São Luís 65080-805, MA, Brazil

**Keywords:** gingival treatment, low-level light therapy, periodontal guided tissue regeneration, gingival recession, oral surgery, connective tissues, tissue grafting, periodontics

## Abstract

Background: Gingival recession (GR) is described as an apical displacement of the gingival margin in relation to the cementoenamel junction, exposing the root surface to the oral cavity environment. This study aimed to evaluate the clinical results of a bilateral root coverage (RC) of GR associated with an autogenous connective tissue graft (aCTG) alone or combined with low-level laser therapy (aCTG + LLLT). Methods: This cross-sectional, split-mouth, double-blind, clinical pilot study featured three individuals who attended a periodontics post-graduate program with the main complaint of GR and dental hypersensitivity (DHS). Of these, only one patient met the inclusion criteria and the parameters evaluated were: DHS, the keratinized tissue’s thickness and width clinical attachment level (CAL), probing on depth (PD), and bilateral GR based on Cairo RT I. The patient was evaluated by a first clinical evaluator and the treatment was randomly divided into two groups, G1: aCTG only (control group, *n* = 3 teeth per side) and G2: aCTG + LLLT (test group, *n* = 3 teeth per side). LLLT used a diode laser (660 nm) with a dose of 3 J/cm^2^ per point and 4 s per point was applied in four different periods, preoperatively; transoperatively and immediately postoperatively, the application was performed in three points (eight applications) on alternate days for 7 days and a 90-day follow-up was performed for clinical evaluations of the periodontal parameters and the collected data were analyzed by Kruskal–Wallis and Dunn tests. Results: the RC mean percentage was <95% in both groups after 90 days. Comparing treatment sides, G1 (*n* = 3/3, 100%) had a higher prevalence of RC than G2 had (*n* = 3/3, 95%). DHS significantly decreased after 90 days in both groups. Both groups showed an improvement in the other periodontal parameters evaluated during the short-term follow-up; mainly, PD had a statistically significant (*p* ˂ 0.05) increase after 90 days and a CAL decrease during this period; KTW and KTT also had a significant increase in both groups (*p* ˂ 0.05). Conclusions: the results indicated that aCTG + LLLT might have an additional benefit to GR root coverage within the evaluated time and this section also includes the within-study limitations.

## 1. Introduction

Low-level laser therapy (LLLT) was first described in the literature by Mester et al. [1] as a therapy based on a light amplification by the stimulated emission of radiation and it was used as a therapeutic method in several areas of medicine and dentistry [2,3,4,5]. Basically, LLLT induces intracellular metabolic alterations, resulting in a faster mitosis, fibroblasts migration, and cell matrix overproduction, which demonstrates a major potential for periodontal biostimulation in order to accelerate tissue healing, epithelial biostimulation, and tissue regeneration [6]. 

In addition, LLLT can produce periodontal tissue decontamination through antimicrobial photodynamic therapy (aPDT), which has been defined as microbial eradication by light associated with a non-toxic photosensitizing agent (methylene blue) that targets bacterial cells; thus, for decontamination, this association is mandatory because LLLT by itself has no antimicrobial potential [7]. aPDT might be added like probiotics and ozone therapy as scaling and root planning (SRP) adjunct treatment mainly because, according to a new periodontal classification, there has been several changes in periodontal pathology, especially in periodontal microbiota after the COVID-19 pandemic; therefore, a new adjunct treatment must be sought [8,9].

In dentistry, LLLT is being used in short-term results associated with regular protocol procedures as scaling and root planning to obtain satisfactory clinical results, although the literature shows that currently the LLLT protocol showed only short-term additional benefits in different procedure types [10]. Regarding periodontics, LLLT has been used in non-surgical therapy, SRP, as an adjunct treatment to improve collagen synthesis, fibroblasts and keratinocytes proliferation, maturation and fixation, growth factors increased levels, anti-inflammatory mediators’ liberation, and vascular cells neoformation that will improve wound healing, decrease edema, pain relief, tissue repairing, and increase the odds to gain a clinical attachment level (CAL) [11,12].

As mentioned previously, the literature has demonstrated the effectiveness of LLLT as an adjuvant therapy in non-surgical procedures such as SRP, however, its effectiveness regarding periodontal surgical treatments such as root coverage (RC) surgeries remains uncertain [12]. RC is a coronal sliding flap to increase the gingival height to treat gingival recession (GR) aiming to control root exposure and dental hypersensitivity (DHS); often, it needs to be associated with RC techniques to soft tissue grafts in order to improve the gingival phenotype and increase the keratinized tissue volume [13,14,15]. Additionally, despite a wide variety of RC techniques available to improve the RC clinical results, traditional mucogingival surgery therapy has its limitations, such as edema and postoperative discomfort, mainly when an autogenous connective tissue graft (aCTG) is associated [16,17].

According to Fernandes-Dias et al. [18] and Santamaria et al. [19], LLLT should be used during RC treatment based on the ability of a low-intensity laser to accelerate the wound healing process, promote neovascularization, eliminate DHS, analgesia, edema reduction, decrease postoperative morbidity or discomfort, and increase anti-inflammatory mediators’ liberation and collagen production, which might upgrade the RC results, increasing the patients’ life quality. In theory, photobiomodulation is the ability to apply light to periodontal soft tissue, inducing a mitochondrial photochemical stimulation, that will produce major energy in adenosine triphosphate (ATP) form [20,21].

According to the literature, photobiomodulation can be achieved through using a low-intensity laser in the infrared range with a wavelength between 660 and 980 nm and 0.05 and 0.5 watts for a postoperative surgical healing improvement and pain relief [20,21,22,23]. However, so far, the literature is scarce of studies that evaluates LLLT associated with RC and aCTG, so a possible hypothesis is to use LLLT in order to the improve CAL, probing on depth (PD), keratinized tissue thickness’ gain, and the elimination of DHS. Therefore, the aim of this study was to evaluate the clinical results of bilateral RC of GR associated with autogenous connective tissue graft (aCTG) alone or combined with low-level laser therapy (aCTG + LLLT).

## 2. Materials and Methods

### 2.1. Study Design and Ethical Aspects

This cross-sectional, short-term (90 days), split-mouth, double-blind, clinical pilot study was conducted between January 2022 and June 2022. Each participant was informed about the study’s objectives, design, risks, and benefits. After agreeing to participate and signing the written consent form, the individuals answered an epidemiological and clinical questionnaire using the form of a confidential structured face-to-face interview. This study was conducted in accordance with the Declaration of Helsinki and was approved by the Committee for Ethics in Research of the Research of the Institute of Biological Sciences at the Federal University of Pará under protocol number 2.601.161. 

### 2.2. Study Population and Sample Size

Initially, from all individuals who attended a monthly post-graduation program in periodontics during the study period, only 3 individuals met the inclusion criteria; these 3 individuals were invited to participate in the study, although only 1 of the individuals agreed to sign the written informed consent and participate in the study. This patient had bilateral maxillary canines and both premolars with buccal GR based on Cairo et al.’s [24] classification as RT I or Miller classification as Class I or II [25]. The inclusion criteria for this study were: (a) bilateral multiple gingival recession on maxillary canines and both premolars, (b) ≥18 years old, (c) visible cementoenamel junction (CEJ), (d) tooth with pulp vitality, (e) active periodontal disease absence, (f) visible dental plaque score ≤20%, (g) absence of any kind of active dental pathology, (h) more than 3 areas with periodontal bleeding on probing, (i) probing depth ≤3 mm in included teeth, (j) Cairo RT I or Miller Class I or II, (k) GR must be on upper canine, 1st pre-molar and 2nd pre-molar, and signing the consent form.

The exclusion criteria were: (a) ≤18 years old, (b) individual with systemic comorbidities that contraindicated the surgical procedure, (c) individual using medications that will interfere with the healing process or that contraindicate the surgical procedure, (d) active smokers, (e) pregnant or lactating women, (f) individual who previously submitted to periodontal surgery in the area of interest, (g) an individual with active periodontal disease, (h) teeth with non-carious lesions compromising the CEJ, and an individual with ongoing orthodontic therapy.

At first, the patient was evaluated by calibrated periodontist #1, with previous experience in periodontal clinical studies, and undertook the protocol assessment, which was established by Hefti and Preshaw [26]. This periodontist #1 was not informed about the surgical and LLLT protocols and informed the participant about GR etiology and treatment. Then, the patient was included in a pre-treatment program for oral health and periodontal adequacy by periodontist #1 to eliminate any possible etiological factors related to GR etiology, such as oral hygiene instructions and a non-traumatic brushing technique with a soft brush.

### 2.3. Periodontal and Clinical Parameters Evaluation

Subsequently, as the pre-treatment program, periodontist #1 carried out the epidemiological questionnaire with self-reported answers (age, gender, ethnicity, source, educational level, marital status, presence of dental hypersensitivity, discomfort level of hypersensitivity, toothbrush type, whether oral hygiene is performed, and whether they use dental floss). The DHS diagnosis occurred based on the patient’s self-declared filling out of two different types of pain scale during anamnesis. Among the DHS queries there were (i) pain during eating (acidic or cold foods or drinks); (ii) an exposed dentin surface on the sensitive tooth; (iii) the presence of a parafunctional habit; (iv) and the absence of any dental pathology that could explain the pain.

The Numeric Pain Rating Scale (NPRS) consists of a scale numbered in ascending order from 0 to 10, where 0 means no pain and 10 means more severe pain, and the participant in the self-assessment marked the number that represented the DHS level. The Faces Pain Scale-Revised (FPS-R) consists of a scale of a set of facial expression drawings that illustrate each stage of the pain intensity and the patient was asked to choose the face that best represented the DHS pain level [27].

After that, periodontist #2, who has previous experience in periodontal clinical studies and went through the protocol assessment established by Hefti and Preshaw [26], was responsible for the evaluation of the periodontal clinical parameters and was not informed about the surgical and LLLT protocols. The periodontal and clinical parameters were a visible plaque index (VPI) measured by the simplified oral hygiene index (SOHI); the presence or absence of bleeding on probing (BoP) at the periodontal sites selected for the study; the probing depth (PD) measured in millimeters with a North Carolina periodontal probe (Hu-Friedy, Rio de Janeiro, Brazil); the clinical attachment level (CAL) measured from the CEJ and the bottom of the periodontal pocket; gingival recession (GR) measured from the CEJ to the gingival margin; and the keratinized tissue’s width (KTW) and the keratinized tissue’s thickness (KTT) being measured by a caliper. At last, the surgical procedures were only allowed after an adequate biofilm control liberated by periodontist #2 [18,19].

### 2.4. Allocation and Blinding

In order to improve the procedure blinding, an external researcher who was unaware of the patient recruitment, assessment, and treatment generated an allocation sequence using a computer program to select which hemiarches would be the test and which would be the controls. The sequence was placed and sealed in opaque envelopes that contained the allocation of sides and only the periodontist #3 who would be the surgeon responsible for the RC procedures. After the allocation, the teeth were divided in 2 groups, G1: aCTG only (control group, *n* = 3 teeth per side) and G2: aCTG + LLLT (test group, *n* = 3 teeth per side). Additionally, to minimize the possibility of bias, even the patients were blinded to know which side received the treatment choose by the allocation. 

### 2.5. Periodontal Surgical and Post-Operative Treatment

To perform the surgical procedures and LLLT protocols, the calibrated periodontist #3, who had previous experience in periodontal clinical studies and went through the protocol assessment established by Hefti and Preshaw [25], knew about the groups division, which group would receive LLLT, and the surgical technique. Bilateral GR through 13, 14, and 15 (Figure 1A) and 23, 24, and 25 (Figure 1B) teeth were treated only with aCTG removed from the patient’s hard palate at the same surgical time, with the coronally advanced trapezoidal flap technique developed by Zucchelli and de Sanctis [28]. The bilateral GR were both treated at the same surgery time and during anesthesia; periodontist #3 would receive the allocation results.

In summary, after local anesthesia with 4% articaine and 1,100,000 epinephrine (Nova DFL, Rio de Janeiro, Brazil), the RC procedure was performed with trapezoidal intrasulcular and parapapillary incisions with 15c blades (Swann Morton, Sheffield, UK), followed by subperiosteal tissues divulsion with Buser detacher and periodontal tunnelers (Hu-Friedy, Rio de Janeiro, Brazil) in order to generate the flap mobility. After the flap’s elevation, the root surface received SRP and root surface decontamination, then bilateral aCTG were removed from the hard palate area and sutured in the recipient area of 13, 14, and 15 (Figure 2A) and 23, 24, and 25 (Figure 2B) teeth with non-absorbable 5.0 sutures (Ethicon, Cincinnati, OH, USA).

After the aCTG was completely stabilized, both flaps were coronally advanced and then sutured by non-absorbable 5.0 sutures (Ethicon, Cincinnati, OH, USA) (Figure 3A,B). The participant received postoperative instructions, such as to stop brushing on surgical sites for 14 days after surgery, use 0.12% chlorhexidine rinses three times a day for 14 days, and to postoperatively control the pain with drugs such as 500 mg of sodium dipyrone 6/6 h for 5 days. Non-absorbable sutures were removed after 14 days.

### 2.6. Laser Protocol

The surgical side chosen for the allocation received laser application protocol with a GaAlAs diode laser (Therapy XT DMC, São Paulo, Brazil). The laser application was performed during 5 different periods of the surgical procedure; the first irradiation was in the preoperative period on the buccal region and it was carried out in three points on each tooth with a wavelength of 660 nm, an output power of 30 mW, a power density of 15 J/cm^2^ (3 J/cm^2^ per point), and it was delivered by application in the continuous wave mode for 4 s [18,19]. During all laser irradiations, the laser tip was placed a slightly safe distance from the gingival tissue at a perpendicular angle due to blood or fluid contact to prevent the contamination of the laser tip. 

In the course of the surgical procedure, a second irradiation was conducted after the evaluation of the. The laser application was performed in three points directly to the buccal periodontal periosteum of the teeth involved with a wavelength of 808 nm, an output power of 30 mW, a power density of 15 J/cm^2^ (3 J/cm^2^ per point), and it was delivered by an application in a continuous wave mode for 4 s. Then, the third irradiation took place after removing aCTG from the hard palate and before suturing it in the buccal periodontal periosteum. The aCTG laser application was carried out in three points throughout aCTG length with a wavelength of 660 nm, an output power of 30 mW, a power density of 15 J/cm^2^ (3 J/cm^2^ per point), and it was delivered by application in the continuous wave mode for 4 s [18,19].

A fourth irradiation happened during the immediate postoperative period with a wavelength of 808 nm, an output power of 30 mW, a power density of 15 J/cm^2^ (3 J/cm^2^ per point), and it was delivered by an application in a continuous wave mode for 4 s. Subsequently, during the postoperative recovery, there was a wavelength of 660 nm, an output power of 30 mW, a power density of 15 J/cm^2^ (3 J/cm^2^ per point), and it was delivered by an application in a continuous wave mode for 4 s [18,19]. 

### 2.7. Statistical Analysis

Descriptive statistics were expressed as the mean ± standard deviation. The frequency of RC, KTW, and KTT increases and the DHS improvement was compared using the Kruskal–Wallis and Dunn’s test with a significance level of 0.05.

## 3. Results

### 3.1. Sample Sociodemographic and Clinical Characteristics

G1 was composed of 23, 24, and 25 and G2 was composed of 13, 14, and 15. Table 1 show the sociodemographic and clinical data demonstrating the comparison between the groups and during the pre-treatment. The postoperative follow-up went through 90 days, with the baseline starting at the immediate postoperative period, reassessed at the 90th day, and it currently remains in a periodontal supportive therapy protocol, as showed in Figure 4A,B, which was performed by periodontist #3 only. 

### 3.2. Dental Hypersensitivity

During the postoperative follow-up, the DHS had decreased in the immediate postoperative period of up to 90 days. G2 received the aCTG + LLLT protocol and the patient self-declared in the NPRS and FPS-R analysis a 100% absence of pain or any kind of DHS on teeth 13, 14, and 15 during feeding or toothbrushing. However, in G1, which received the aCTG protocol, the patient self-declared in the NPRS and FPS-R analysis a 100% absence of pain or any kind of DHS on teeth 23 and 25, but on tooth 24, the patient self-declared through the iconography and number scale that they felt mild pain while eating, mainly when eating cold drinks or foods. Comparing the patient’s initial status 90 days, they had a 90% improvement.

### 3.3. Dental Biofilm Control

For dental plaque control, SOHI was performed from the immediate postoperative period up to 90 days; the patient demonstrated a 100% improvement in the dental biofilm control. Initially, the patient’s biofilm accumulation was rated by SOHI below 20% in both groups during the follow-up period, demonstrating the effective control of supragingival plaque. Regarding traumatic toothbrushing with a hard-bristle toothbrush, after an initial explanation about its damage to the periodontium, the patient changed to a soft-bristle toothbrush, resulting in a complete absence of harmful habits.

### 3.4. Probing Depth and Clinical Attachment Level

The 90-day evaluation showed that in both groups, there was an increase in the PD and decrease in the CAL. The PD mean values were initially 1.66 ± 0.51 mm for both groups, but, in the final evaluation, the G1 mean values were 3.00 ± 0.00 mm and the G2 mean values were 2.66 ± 0.57 mm. The difference between the groups was statistically significant (*p* = 0.018) and when we analyzed the groups for the PD increase, according Table 2, there was a statistical significance (*p* ˂ 0.05). Regarding the CAL, in the final evaluation, the G1 mean values were 3.00 ± 0.00 mm and the G2 mean values were 2.66 ± 0.57 mm after 90 days and when comparing the CAL initial mean values of 4.16 ± 0.40 with the 90-day results, there was not a significant difference (*p* = 0.07) (Table 2).

### 3.5. Gingival Recession

Initially, both groups showed mean a GR of 2.50 ± 0.54 mm and after 90 days, both groups demonstrated a statistically significant reduction in the GR. After 90 days, G2 had 100% RC rates and G1 had tooth 23 with only a 90% of coverage of the exposed area, yet teeth 24 and 25 showed root coverage percentages of 95%, respectively, meaning there was an average reduction of 1.00 ± 0.00 and the RC mean percentage was <95% in both groups after 90 days. Although G1 had lower RC rates than G2, there was not a statistical significance between the covering differences because the exposed area was less than 1 mm.

### 3.6. Keratinized Tissue Evaluation

In concern of the keratinized tissue’s thickness and width, there was a significant increase in both groups (*p* ˂ 0.05). The initial KTW presented an average of 1.00 ± 0.00 and after treatment, regardless of the group treatment, both had an average gain of 3.00 ± 0.00. The initial KTT had an average of 1.16 ± 0.40 and after treatment, regardless of the group treatment, both had an average gain of 2.00 ± 0.00 in both groups and after 90 days, there was no significant difference between the groups (*p* = 0.019).

## 4. Discussion

It is a worldwide consensus on mucogingival plastic procedures that in order to improve RC clinical results, there is a need to associate with aCTG, in single or multiple GR cases, mainly using Miller Class I and II types or Cairo RT I recessions which, according to the literature, has a 100% possibility of RC. However, upon the development of new technologies, there are new possibilities to improve the RC clinical results in the short-term period and long-term period. Based on new technologies, the use of LLLT as an adjuvant therapy in periodontal treatment has been growing and new indications must be developed in both non-surgical and surgical procedures.

Among these new indications for LLLT in periodontal surgical procedures in the short-term is the decrease of postoperative pain and edema, the improvement of wound healing, especially in the hard palate region after aCTG removal, and the acceleration of collagen production. According to Zhao et al. [29], there is a significant difference in post-surgical pain relief when LLLT is used. In the 3rd postoperative day, however, and also the 7th postoperative day, no difference was observed. From the 14th postoperative day onwards, a faster re-epithelialization and an improvement in the healing of the palatal donor area was seen.

Despite the literature promising LLLT results regarding the patients’ morbidity in the postoperative period, mainly in aCTG removal donor areas, it is insufficient to support the effectiveness of LLLT in periodontal clinical outcomes such as a decrease in the CAL and GR, as well as the enlargement of KTW and KTT [23]. In addition to the insufficient data, the variance in the laser protocols among studies that use aCTG + LLLT may impair the evaluation of the effectiveness. Exemplifying this problem, Ustaoglu et al. [30], Lafzi et al. [20], and Morshedzadeh et al. [31] evaluated the effectiveness of LLLT associated with an autogenous free gingival graft in a short-term period to improve the periodontal clinical results, however, no statistically significant differences were found between the groups of those three studies in terms of an improvement in the CAL or PD.

Based on this high heterogeneity of the methodologies and different techniques to regenerate periodontal tissue, a question still remains unanswered regarding the LLLT capability to improve the CAL, PD, or KTT when associated with periodontal surgeries. This pilot study aimed to evaluate in a short-term period the RC results associated with aCTG + LLLT to treat bilateral multiple GR. Our previous results indicate that LLLT can increase the aCTG predictability of an improvement in the CAL, PD, or KTT in our patient.

Comparing the groups’ data, G2 had a 100% RC rate, while G1 had 90% and 95% coverage rates. Possibly, these slight differences between the coverage rates could have occurred due to the phenomenon of photobiomodulation caused by the lasers’ visible wavelength in the periosteum tissues to improve wound healing, promote epithelial cells proliferation, and fibroblasts and collagen deposition that, in a short-term period, may have an influence on the coverage rates [32]. In a long-term period, both groups irradiated with LLLT could achieve the total coverage aided by *Creeping Attachment* phenomenon, as described by Pini-Prato et al. [33] and Pereira Neto et al. [34], depending on the region treated and the tissue graft’s type.

Our results confirmed the findings of previous studies, such as those of Fernandes-Dias et al. [16], Santamaria et al. [17], and Ozturan et al. [32], that described a positive association between low-power laser biostimulation as an adjuvant therapy and a coronally advanced flap with an autogenous connective tissue graft for RC. Ozturan et al. [32] reported statistically significant differences between the test and control groups regarding the improvement in the CAL, PD, KTW, and KTT parameters; the test group had a total RC as reported in our study. Ozturan et al. [32] also used a diode laser with a wavelength of 588 nm and a power density of 4 J/cm^2^ after a coronal advanced flap; however, the authors did not cite the time used, the application mode, or the output power.

In our study, a diode laser was used in two different protocols based on previous studies; the first protocol had a wavelength of 660 nm, an output power of 30 mW, and a power density of 15 J/cm^2^ (3 J/cm^2^ per point) for 4 s in a continuous wave mode at the hard palate site, the periodontal flap site, and the aCTG out of the mouth. The second protocol used a wavelength with an 808 nm output power of 30 mW and a power density of 15 J/cm^2^ (3 J/cm^2^ per point) for 4 s in the continuous wave mode, directly in the bilateral periosteum. Studies have shown that power densities of 3–6 J/cm^2^ are capable of photobiostimulate wound healing, fibroblast proliferation, collagen production, and have anti-inflammatory activity [18,19]. However, unlike the literature, our study used LLLT both in the removed aCTG and periosteum in order to improve the PD, KTW, and KTT parameters, which also induced a decrease in the CAL [33].

However, we need to show caution when interpreting our data. Because the bilateral and multiple GR used in both groups were shallow recessions (2–3 mm), this is a possible bias to our study because it has a better prognosis to a 100% probability of coverage, especially when it was treated with the Zucchelli and de Sanctis [28] RC technique. Furthermore, the literature shows a considerable results variation in relation to aCTG and LLLT to treating GR; Fernandes-Dias et al. [18], Santamaria et al. [19], and Ozturan et al. [32] corroborated that there is a minimal improvement in the periodontal clinical parameters during a short-term period. Although, Lavu et al. [21] demonstrated that for aCTG + LLLT, there is significant improvement in the periodontal clinical parameters, which is different from our results due to the achievement of 100% coverage rates on G2; this result is promising when compared with Fernandes-Dias et al. [18] who obtained 65% coverage rates, but there is a significant difference between the sample sizes.

Regarding DHS, few studies evaluated the DHS after RC procedures with aCTG + LLLT and none with our methodology. The presence of DHS was assessed before, during, and after the periodontal treatment and was always self-declared by the patient to avoid any kind of bias. Two different pain scales were used to confirm the presence of DHS and its improvement; among our results, we had a statistically significant reduction in the DHS during the periodontal treatment and after 90 days; however, no statistically significant difference was observed between the test and the control groups after 90 days. Based on our results and based on the literature, it can be inferred that surgical procedures such as RC can reduce DHS and improve the patient’s quality of life [35]. 

Within the study’s limitations, we can highlight the low sample size due to the short period of the individuals’ recruitment, the short postoperative evaluation period, the easily predictable GR types for a total coverage—possibly other types should be evaluated in another study—and the variability in the methodology to the total coverage compared to the literature. However, this pilot study was important to show the need to use minimally invasive approaches or systems, as demonstrated by Scribante et al. [36]. Although, we need a bigger sample size, to increase the evaluation period, and evaluate the possibility of using used hyaluronic acid as complete LLLT in order to improve the RC over a long time period.

## 5. Conclusions

We concluded that the present study can contribute to answering questions about the effectiveness of LLLT on periodontal soft tissues during surgical procedures, although, within this study, limitations were found that the LLLT may bring minimal additional benefits in the total root coverage associated with an autogenous connective tissue graft in a short-term period among Cairo RT I multiple recessions. However, more studies are needed to obtain more conclusive feedback. Although, based on our results, LLLT might deserve to be considered for these results and its ability to be applied in several clinical fields. 

## Figures and Tables

**Figure 1 jcm-12-02349-f001:**
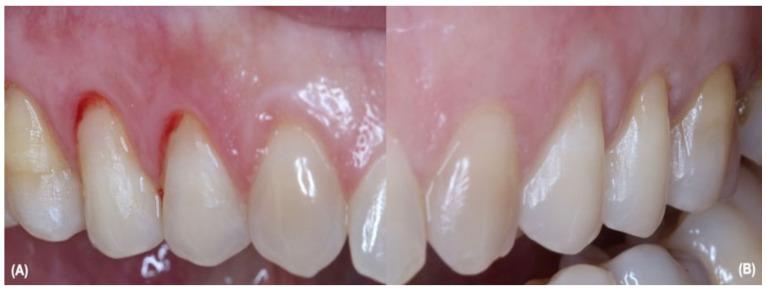
(**A**) shows multiple GR on teeth 13, 14, and 15; (**B**) show multiple GR on teeth 23, 24, and 25.

**Figure 2 jcm-12-02349-f002:**
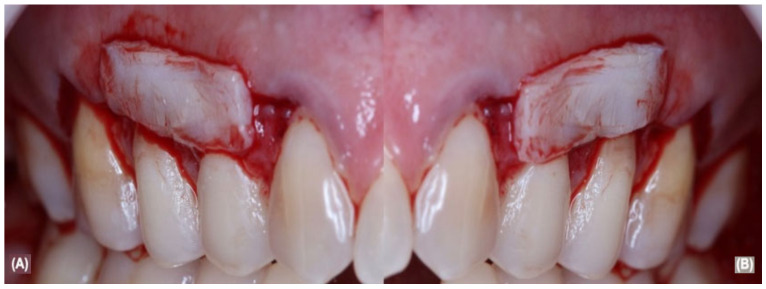
(**A**) shows flap elevated with ACTG on top of right periodontium; (**B**) show flap elevated with ACTG on top of left periodontium.

**Figure 3 jcm-12-02349-f003:**
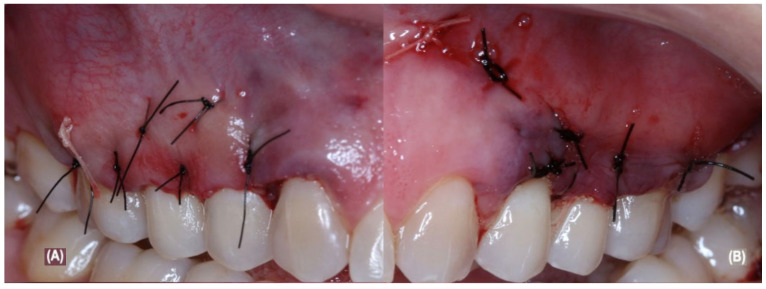
(**A**) show right flap sutured; (**B**) show left flap sutured.

**Figure 4 jcm-12-02349-f004:**
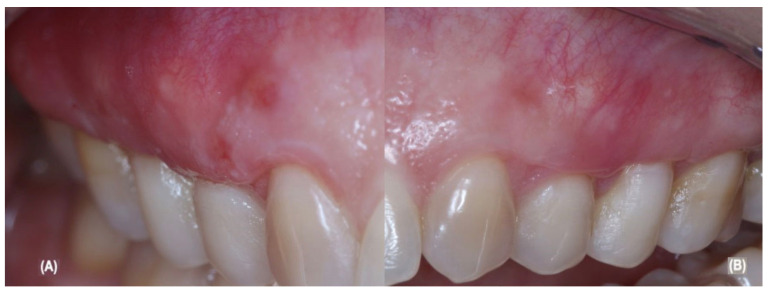
(**A**) show aCTG + LLLT 90-day follow-up; (**B**) show aCTG 90-day follow-up.

**Table 1 jcm-12-02349-t001:** Socio-epidemiological and clinical data at baseline and after 90 days.

Parameters	Pre-Treatment	G1: aCTG Treatment	G2: aCTG + LLLT Treatment
Age (years)	35 years	-	-
Sex	Female	-	-
Ethnicity ^†^	White	-	-
Source	Belém	-	-
Educational level	Completed post-graduation	-	-
Marital status	Married	-	-
Traumatic brushing	Yes	No	No
Dental hypersensitivity	Yes	No	No
Pain during feeding	Yes	No	No
Exposed dentin root surface	Yes	No	No
Parafunctional habit	No	No	No
Numerical scale of hypersensitivity	Level 7	Level 1	Level 0
Facial pain scale	Moderate pain iconography	Light pain iconography	No pain iconography
Toothbrush type	Hard bristles	Soft bristles	Soft bristles
Oral Hygiene	Yes	Yes	Yes
Dental floss use	Yes	Yes	Yes
Simplified oral hygiene index	Level 1	Level 0	Level 0
Bleeding on probing *	Absent	Absent	Absent
Probing depth *			
Tooth 13	1 mm	-	2 mm
Tooth 14	2 mm	-	3 mm
Tooth 15	2 mm	-	3 mm
Tooth 23	1 mm	3 mm	-
Tooth 24	2 mm	3 mm	-
Tooth 25	2 mm	3 mm	-
Clinical attachment level *			
Tooth 13	4 mm	-	2 mm
Tooth 14	4 mm	-	3 mm
Tooth 15	5 mm	-	3 mm
Tooth 23	4 mm	3 mm	-
Tooth 24	4 mm	3 mm	-
Tooth 25	4 mm	3 mm	-
Gingival recession *			
Tooth 13	3 mm	-	Absent
Tooth 14	2 mm	-	Absent
Tooth 15	3 mm	-	Absent
Tooth 23	3 mm	1 mm	-
Tooth 24	2 mm	≤1 mm	-
Tooth 25	2 mm	≤1 mm	-
Keratinized tissue width *			
Tooth 13	1 mm	-	3 mm
Tooth 14	1 mm	-	3 mm
Tooth 15	1 mm	-	3 mm
Tooth 23	1 mm	3 mm	-
Tooth 24	1 mm	3 mm	-
Tooth 25	1 mm	3 mm	-
Keratinized tissue thickness *			
Tooth 13	1 mm	-	2 mm
Tooth 14	1 mm	-	2 mm
Tooth 15	2 mm	-	2 mm
Tooth 23	1 mm	2 mm	-
Tooth 24	1 mm	2 mm	-
Tooth 25	1 mm	2 mm	-

* Evaluation made per tooth; ^†^ Self-declared.

**Table 2 jcm-12-02349-t002:** Mean and standard deviation analysis at baseline and after 90 days.

Parameters (Measure Unit)	Pre-Treatment	G1: aCTG * Treatment	G2: aCTG + LLLT ^†^ Treatment
Probing depth (mm)	1.66 ± 0.51	3.00 ± 0.00	2.66 ± 0.57
Clinical attachment level (mm)	4.16 ± 0.40	3.00 ± 0.00	2.66 ± 0.57
Gingival recession (mm)	2.50 ± 0.54	1.00 ± 0.00	-
Keratinized tissue width (mm)	1.00 ± 0.00	3.00 ± 0.00	3.00 ± 0.00
Keratinized tissue thickness (mm)	1.16 ± 0.40	2.00 ± 0.00	2.00 ± 0.00

* Autogenous connective tissue graft (aCTG); ^†^ Autogenous connective tissue graft + low-level laser therapy.

## Data Availability

The data are available upon request from the corresponding author.

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
