# Peer review of "Clinical Evaluation of Bilateral Multiple Gingival Recession Treatment with Autogenous Connective Tissue Graft Associated with Low-Level Laser Therapy"

_jcm, 2023, doi:10.3390/jcm12062349_

Round 1

Reviewer 1 Report

Title: Clinical evaluation of bilateral multiple gingival recession treatment with autogenous connective tissue graft associated with low-level laser therapy.

Manuscript ID: jcm-2160385

The application of different types of laser in Periodonics is a novel line of research. Aspects of improvement to the authors for their manuscript are proposed below.

Key Words

-Use appropriate and correct MeSH terms as keywords, so that if the manuscript is accepted, it appears properly in the bibliographical repertoires (PubMed).

Introduction

-At the end of the first paragraph, the authors state: "LLLT can produce periodontal tissue decontamination through antimicrobial photodynamic therapy (aPDT)....". PDT is a non-surgical approach dependent on light, oxygen, and photosensitizing agents. LLLT uses a photosensitizer?. Clarify.

- Second paragraph. The meta-analysis by Ren C et al. (reference 9), shows that LLLT failed to show significant additional intermediate-term (3 and 6 mo) effects in terms of clinical parameters and alveolar bone density. These findings indicated that LLLT showed only short-term additional benefits after conventional SRP. Reference 10 is a narrative review that insists on the previous statements. Therefore, the authors must be precise and cautious in their statements. This paragraph must be nuanced or worded differently.

M&M

- 2.1. A cross sectional study or an RCT?. If it is a clinical trial, how many items on the CONSORT checklist did the study meet?.

-2.2. The inclusion criteria should be clarified. Are they referring only to the teeth that were the reason for the intervention, or to the entire dentition?. The inclusion criterion h is not understood, what are areas? If the patient is not diagnosed with periodontitis, does he have gingivitis?...... Clarify.

- If the study only includes one patient, would it be more correct to speak of "a clinical case" and not an epidemiological study? If this is correct, the authors believe that it should be described in the title. Clarify.

- Authors speak several times in the text of patients in plural. Correct.

- The authors say: "Then, the patient was included in a pre-treatment program by periodontist #1 to eliminate possible etiological factors related to GR etiology, ......... and SRP." If the patient is not periodontal, what do the authors understand by SRP treatment?.

2.3. What does it consist of and how were the KTW and KTT variables measured?.

2.4. A single patient with a split mouth design, was a computer program necessary to randomly designate a sector?.

2.5. Periodontists 1 and 3 were calibrated. With whom, what concordance figures, Kappa or correlations did they present?.

- The headings of figures 2 and 3 do not correspond. Correct.

Results

- The authors state: "90 days evaluation showed that in both groups an increase of PD and CAL." Table 2 shows the opposite in CAL, a reduction in measurements.

- Table 2. Units of measure are missing.

- PD mean difference between G1 and G2 was not statistically significant (p=0.018). This significance value IS statistically significant. Clarify.

-Although when we analyzed groups for PD decrease, there was statistical significance (p Ë‚ 0.05)”. The PD values increase with respect to the baseline, according to Table 2. Correct.

- The authors repeat in writing all the results of the periodontal variables in Table 2. What sense does it make to include the table? Information is being repeated. My advice is to complete Table 2 with units of measurement, include the p values and fuse the most notable periodontal results into a single section in the text.

Discussion

- Line 324. "LLLT in periodontal clinical outcomes such as CAL increase, GR decrease...." Do you mean CA gain?.

- Line 337.The authors' study is not done on a sample, but on a patient. Correct.

Author Response

Reply to reviewer #1

1. Concern of the reviewer • Key Words - Use appropriate and correct MeSH terms as keywords, so that if the manuscript is accepted, it appears properly in the bibliographical repertoires (PubMed). 

Our response: Dear Reviewer #1, we appreciate your suggestion and the keywords were carefully revised according pubmed mesh database.

Revised text:Page 1, lines 45-46, “From January 1999 to January 2013, 161 individuals were informed about the purpose of the study and invited to participate, only 113 agreed and signed written consent form before data collection and dental evaluation.” 

2. Concern of the reviewer

• Introduction - At the end of the first paragraph, the authors state: "LLLT can produce periodontal tissue decontamination through antimicrobial photodynamic therapy (aPDT)....". PDT is a non-surgical approach dependent on light, oxygen, and photosensitizing agents. LLLT uses a photosensitizer?. Clarify.  

Our response: Dear Reviewer #1, we appreciate your concern and the text was carefully explained to readers. 

Revised text:Page 2, lines 58-59, “so for decontamination this association is mandatory because LLLT by itself has no antimicrobial potential” 

3. Concern of the reviewer

• Introduction - The meta-analysis by Ren C et al. (reference 9), shows that LLLT failed to show significant additional intermediate-term (3 and 6 mo) effects in terms of clinical parameters and alveolar bone density. These findings indicated that LLLT showed only short-term additional benefits after conventional SRP. Reference 10 is a narrative review that insists on the previous statements. Therefore, the authors must be precise and cautious in their statements. This paragraph must be nuanced or worded differently. 

Our response: Dear Reviewer #1, we appreciate your suggestion and the text was carefully explained. However, the cited references were used to describe LLLT tissue and cell mechanism as seen in paragraph. 

Revised text:Page 2, lines 60-63, “in short-term results associated to regular protocol procedures as scaling and root planning to obtain satisfactory clinical results, although literature shows that currently LLLT protocol showed only short-term additional benefits in different procedure types.”   

4. Concern of the reviewer

• M&M - 2.1. A cross sectional study or an RCT?. If it is a clinical trial, how many items on the CONSORT checklist did the study meet?. 

Our response: Dear Reviewer #1, we appreciate your suggestion and the text was carefully revised as cross sectional study.  

5. Concern of the reviewer

• M&M - 2.2. The inclusion criteria should be clarified. Are they referring only to the teeth that were the reason for the intervention, or to the entire dentition?. The inclusion criterion h is not understood, what are areas? If the patient is not diagnosed with periodontitis, does he have gingivitis?...... Clarify. 

Our response: Dear Reviewer #1, we appreciate your suggestion and the text was carefully revised. 

Revised text:Page 3, lines 111-124, “The inclusion criteria for this study were: a) bilateral multiple gingival recession on maxillary canines and both premolars, b) ≥ 18 years old, c) visible cementoenamel junction (CEJ), d) tooth with pulp vitality, e) active periodontal disease absence, f) visible dental plaque score ≤20%, g) absence of any kind of active dental pathology, h) more than 3 areas with periodontal bleeding on probing, i) probing depth ≤ 3 mm in included teeth (j) Cairo RT I or Miller Class I or II, (l) GR must be on upper canine, 1st pre-molar and 2nd pre-molar and patients who signed the consent form. Exclusion criteria were: a) ≤ 18 years old, b) patients with systemic comorbidities that contraindicated the surgical procedure, c) patients using medications that will interfere with the healing process or that contraindicate the surgical procedure, d) active smokers, e) pregnant or lactating women, f) patients who previously submitted to periodontal surgery in the area of ​​interest, g) patients with active periodontal disease, h) teeth with non-carious lesions compromising the CEJ and patients with ongoing orthodontic therapy.” 

6. Concern of the reviewer

• M&M - If the study only includes one patient, would it be more correct to speak of "a clinical case" and not an epidemiological study? If this is correct, the authors believe that it should be described in the title. Clarify 

Our response: Dear Reviewer #1, we appreciate your suggestion and we suggest to read the revised text which we explained about the presence of only one patient in the study.  

Revised text:Page 3, lines 107-110, “only 3 individuals met the inclusion criteria and these 3 individuals were invited to participate of the study, although only 1 individual invited to the study agreed to sign the written informed consent and participate the study.”    

7. Concern of the reviewer

• M&M - Authors speak several times in the text of patients in plural. Correct. 

Our response: Dear Reviewer #1, we appreciate your suggestion and the term was modified to individual. 

8. Concern of the reviewer

• M&M - The authors say: "Then, the patient was included in a pre-treatment program by periodontist #1 to eliminate possible etiological factors related to GR etiology, ......... and SRP." If the patient is not periodontal, what do the authors understand by SRP treatment?. 

Our response: Dear Reviewer #1, we appreciate your suggestion and the text was carefully revised. 

Revised text:Page 3, lines 129-132, “Then, the patient was included in a pre-treatment program for oral health and periodontal adequacy by periodontist #1 to eliminate possible etiological factors related to GR etiology, such as oral hygiene instructions and non-traumatic brushing technique with a soft brush.” 

9. Concern of the reviewer

• M&M - What does it consist of and how were the KTW and KTT variables measured?. 

Our response: Dear Reviewer #1, we appreciate your suggestion and the text was carefully revised. 

Revised text:Page 3, lines 157-158, “were measured by caliper, at last.” 

10. Concern of the reviewer

• M&M - A single patient with a split mouth design, was a computer program necessary to randomly designate a sector?. 

Our response: Dear Reviewer #1, we appreciate your concern, although as presented in our references all studies involving our theme used a fourth non related to the study person to use a computer program to designate sections and their treatment in order to prevent bias and the text was carefully revised. 

11. Concern of the reviewer

• M&M - Periodontists 1 and 3 were calibrated. With whom, what concordance figures, Kappa or correlations did they present?. 

Our response: Dear Reviewer #1, we appreciate your suggestion, unfortunately we were not able to perform Kappa in this pilot study, it will be done in the mani study, however to prevent any bias all periodontists involved went through periodontal protocol assessment for periodontal research established by Hefti & Preshaw [23]. 

12. Concern of the reviewer

• M&M - The headings of figures 2 and 3 do not correspond. Correct.  

Our response: Dear Reviewer #1, we appreciate your suggestion and all headings were carefully revised. 

13. Concern of the reviewer • Results - The authors state: "90 days evaluation showed that in both groups an increase of PD and CAL." Table 2 shows the opposite in CAL, a reduction in measurements. 

Our response: Dear Reviewer #1, we appreciate your suggestion and the text was carefully revised. 

Revised text:Page 9, lines 284, “decrease of CAL.” 

14. Concern of the reviewer • Results - Table 2. Units of measure are missing. 

Our response: Dear Reviewer #1, we appreciate your suggestion and the text was carefully revised. 

15. Concern of the reviewer

• Results - PD mean difference between G1 and G2 was not statistically significant (p=0.018). This significance value IS statistically significant. Clarify. 

Our response: Dear Reviewer #1, we appreciate your suggestion and the text was carefully revised. 

Revised text:Page 9, lines 285-288, “PD mean values were initially 1.66±0.51mm for both groups, but, in the final evaluation G1 mean values were 3.00±0.00mm and G2 mean values were 2.66±0.57mm and difference between the groups was statistically significant (p=0.018) and when we analyzed groups for PD increase according table 2 with a statistical significance (p Ë‚ 0.05).” 

16. Concern of the reviewer • Results - “Although when we analyzed groups for PD decrease, there was statistical significance (p Ë‚ 0.05)”. The PD values increase with respect to the baseline, according to Table 2. Correct. 

Our response: Dear Reviewer #1, we appreciate your suggestion and the text was carefully revised. 

Revised text:Page 9, lines 285-288, “PD mean values were initially 1.66±0.51mm for both groups, but, in the final evaluation G1 mean values were 3.00±0.00mm and G2 mean values were 2.66±0.57mm and difference between the groups was statistically significant (p=0.018) and when we analyzed groups for PD increase according table 2 with a statistical significance (p Ë‚ 0.05).”

17. Concern of the reviewer

• Results - The authors repeat in writing all the results of the periodontal variables in Table 2. What sense does it make to include the table? Information is being repeated. My advice is to complete Table 2 with units of measurement, include the p values and fuse the most notable periodontal results into a single section in the text. 

Our response: Dear Reviewer #1, we appreciate your suggestion, although we would like to maintain the text in order to help new readers to understand our results.  

18. Concern of the reviewer • Results - Line 324. "LLLT in periodontal clinical outcomes such as CAL increase, GR decrease...." Do you mean CA gain?. 

Our response: Dear Reviewer #1, we appreciate your suggestion and the text was carefully revised. 

Revised text:Page 10, lines 327-328, “such as CAL and GR decrease, also KTW and KTT enlargement.” 

19. Concern of the reviewer • Results - Line 337.The authors' study is not done on a sample, but on a patient. Correct. 

Our response: Dear Reviewer #1, we appreciate your suggestion and the text was carefully revised. 

Revised text:Page 10, line 340, “patient.”

Reviewer 2 Report

manuscript of considerable interest for the dental sector, before proceeding with a possible publication it needs a minor revision

Abstract: highlighting the results

keywords; few, to add specifics

Introduction: add the new classification of periodontal disease focusing on the evolution of the pathology, how the microbiota changes and the use of probiotics and ozone therapy, as already studied by the research group of Prof Scribante

Materials and Methods: Well described

Results: highlight statistically significant data

Discussion: Add the use of hyaluronic acid as future goals.

Conclusions: rewrite by adding a proactive approach

Bibliography: add references required

Author Response

Reply to reviewer #2

1. Concern of the reviewer             

• Abstract: highlighting the results. 

Our response: Dear Reviewer #2, we appreciate your concern. The text was carefully added. 

Revised text:Page 1, lines 38-43, “RC mean percentage was <95% in both groups after 90 days. Comparing treatment sides, G1 (n=3/3, 100%) had higher prevalence of RC than G2 had (n=3/3, 95%). DHS significantly decreased after 90 days in both groups. Both groups showed improvement in the other periodontal parameters evaluated during short term follow-up, mainly, PD had statistical significance (p Ë‚ 0.05) increase after 90 days and CAL decrease during this period, also KTW and KTT which had a significant increase in both groups (p Ë‚ 0.05.” 

2. Concern of the reviewer• keywords; few, to add specifics. 

Our response: Dear Reviewer #2, we appreciate your concern. The text was carefully added.

Revised text:Page 1, lines 46-47, “Low-Level Light Therapy; Periodontal Guided Tissue Regeneration; Gingival Recession; Oral Surgery; Connective Tissues; Tissue Grafting; Periodontics.”

3. Concern of the reviewer

Introduction: add the new classification of periodontal disease focusing on the evolution of the pathology, how the microbiota changes and the use of probiotics and ozone therapy, as already studied by the research group of Prof Scribante.

Our response: Dear Reviewer #2, we appreciate your suggestion. The text was carefully added. 

Revised text:Page 2, lines 60-64,aPDT might be added like probiotics and ozone therapy as scaling and root planning (SRP) adjunct treatment mainly because, according new periodontal classification there has been several changes periodontal pathology, specially in periodontal microbiota after COVID-19, therefore new adjunct treatment must be sought [8,9].”

4 Concern of the reviewer

Materials and Methods: Well described.

Our response: Dear Reviewer #2, we appreciate your kindness.   

5. Concern of the reviewer

Results: highlight statistically significant data.

Our response: Dear Reviewer #2, we appreciate your kindness. 

6. Concern of the reviewer

Discussion: Add the use of hyaluronic acid as future goals.

Our response: Dear Reviewer #2, we appreciate your suggestion. The text was carefully added. 

Revised text:Page 11, lines 401-402,and evaluate the possibility to used hyaluronic acid as complete to LLLT in order to improve RC long period time.”

7. Concern of the reviewer

Conclusions: rewrite by adding a proactive approach.

Our response: Dear Reviewer #2, we appreciate your suggestion. The text was carefully added. 

Revised text:Page 11, lines 408-410,however, more studies are needed to more conclusive feedback, however based on our results LLLT might deserve to be considered for its currently results and applicability in several clinical fields.”

8.Concern of the reviewer

Bibliography: add references required

Our response: Dear Reviewer #2, we appreciate your suggestion. The text was carefully added. 

Revised text:Page 12, lines 442-445,1- Butera A, Maiorani C, Morandini A, Simonini M, Colnaghi A, Morittu S, Barbieri S, et al. Assessment of Oral Microbiome Changes in Healthy and COVID-19-Affected Pregnant Women: A Narrative Review. Microorganisms 2021, 9(11):2385.  2- Colombo M, Gallo S, Garofoli A, Poggio C, Arciola CR, Scribante A. Ozone Gel in Chronic Periodontal Disease: A Randomized Clinical Trial on the Anti-Inflammatory Effects of Ozone Application. Biology (Basel) 2021, 10(7):625.” 

Round 2

Reviewer 1 Report

18 improvement proposals have been corrected, clarified and/or included in the new revised manuscript. Only proposal number 6 has not been adequately addressed. Since the study is based on a single patient, the authors must include in the text, in a subtitle (although it is not the most accurate), a clinical case. The title should reflect, as much as possible, the type of study carried out.